# On the Relationship of Viral Particles and Extracellular Vesicles: Implications for Viral Vector Technology

**DOI:** 10.3390/v13071238

**Published:** 2021-06-26

**Authors:** Christoph Metzner, Marianne Zaruba

**Affiliations:** Institute of Virology, University of Veterinary Medicine, 1210 Vienna, Austria; marianne.zaruba@vetmeduni.ac.at

**Keywords:** extracellular vesicles, exosomes, viral vectors, gene therapy, biomedicine

## Abstract

Gene therapy vectors derived from different viral species have become a fixture in biomedicine, both for direct therapeutic intervention and as tools to facilitate cell-based therapies, such as chimeric antigen receptor-based immunotherapies. On the contrary, extracellular vesicles have only recently gained a massive increase in interest and, concomitantly, knowledge in the field has drastically risen. Viral infections and extracellular vesicle biology overlap in many ways, both with pro- and antiviral outcomes. In this review, we take a closer look at these interactions for the most prominent groups of viral vectors (Adenoviral, Adeno-associated and Retro/Lentiviral vectors) and the possible implications of these overlaps for viral vector technology and its biomedical applications.

## 1. Introduction: Viral Vectors and Extracellular Vesicles

In the last decade, lipid enclosed particles of cellular origin, unable to replicate independent of their cellular source (generally termed extracellular vesicles, EVs) and produced by a wide range of cell types from all kingdoms of life, have attracted significant interest. While earlier believed to be artifacts (“cell dust”) or simple waste products, they are emerging as complex signal transduction vectors with implications for a wealth of physiological and pathological situations. Reviews on the topic are numerous [1,2,3,4,5,6,7,8,9,10]. Extracellular vesicles are produced from eukaryotic, as well as bacterial and archaeal, sources [4]. Transfer of vesicles between cells from different kingdoms is possible and may have provided an evolutionary booster. Different subtypes have been described in various size ranges and with various content types (see Figure 1 and Table 1 for more information). Amongst these, the eukaryotic exosomes seem to carry the strongest promise for biomedical applications. Differentiation between subtypes can be difficult. The immunogenicity of EVs is tunable: from inherently low (when autologous) to ensure safe and efficient delivery, to high by recombinantly displaying antigens in vaccine development strategies [11]. EVs are discussed as a means for enabling gene and drug delivery, vaccine development and novel diagnostic strategies. The latter make use of so-called liquid biopsies. These allow us to gain information on cellular or tissue states from body fluid samples, leading to reduced invasiveness and less mechanical stress at the site of, e.g., a solid tumor [12,13]. Loading of protein to EVs has been achieved, e.g., for catalase to RAW264.7 cell-derived exosomes [14], as well as for cytokines, antibody fragments, RNA binding proteins, vaccine antigens and Cas9 proteins [15]. Targeting of proteins to EV compartments is feasible by using scaffold proteins in fusion proteins [15] or by post-translational modifications [16]. Small molecules, such as the cytostatic taxane paclitaxel, have been loaded into EVs [14] as well as the cytostatic anthracycline doxirubicin [17] or the anti-inflammatory and anti-neoplastic polyphenol curcumin [18]. An interesting aspect of exosomal biomedical application is their platform nature, which allows the implementation of more than one modification on the same physical unit. In one example, exosomes were engineered to display RGD peptide motives (thus targeting αv integrins) on their surfaces and simultaneously be loaded with doxorubicin [17]. In this case, different sites on the EVs are being exploited: soluble elements are delivered to the lumen of EVs and lipophilic factors may be attached to the vesicle membrane. Display of targeting peptides has also been used in conjunction with nucleic acid delivery [19]. EVs were loaded with siRNA by electroporation and for targeting, RVG peptides were attached to the surface via fusion with an exosomal marker protein and were used to deliver siRNA to mouse brains intravenously, indicating the capability of EV delivery systems to cross the blood brain barrier and, specifically, enter neurons, microglia and oligodendrocytes in the brain [19]. Mostly, small non-coding RNA species are used in EV-based delivery strategies [20,21,22]; however, linear DNA was also reported to associate with EVs [23].

EVs are defined as submicron lipid bilayer enclosed vesicles that are unable to replicate independently. This definition includes enveloped virus (eVI) and already hints at the close ties between VIs and EVs [24,25,26,27,28] (see also Table 1 and Figure 2). Introducing nucleic acid to target-cells and—in the case of wild type (wt) virus—replicating and ensuring expression of viral proteins is at the core of viral life cycles. Over time, due to (co-)evolution and selection of the best adapted means of replication in a specific host, a degree of technical “optimization” has been achieved that artificial or synthetic systems are lacking. Not surprisingly, biomedicine is exploiting the fact: amongst others, members of the families *Retroviridae* (i.e., Murine leukemia virus, MLV; Human immunodeficiency virus, HIV), *Adenoviridae* (i.e., adenovirus, AV) and *Parvoviridae* (i.e., adeno-associated viruses, AAV) have become standard tools as viral vector (VV) systems in biotechnology and biomedicine to facilitate high efficiency gene transfer in research and therapeutic settings. Therapies based on the use of VVs may either make direct use by transduction of diseased cells and redressing pathological situations from cancer to infectious diseases, or VVs may be used as tools to enable cell-based therapies, such as chimeric antigen receptor (CAR) based immune-therapies [29,30]. In addition to delivery functions, viral vectors are also used in oncolytic settings: conditional replication of VVs in tumor cells is exploited to directly attack malignancies and mount a more effective anti-tumor immune response [31,32,33,34].

**Table 1 viruses-13-01238-t001:** Overview of Eukaryotic Extracellular Vesicles.

Source	Type	Vesicle	Diameter (in nm)	Density (in g/mL)	Marker	Functions	Reference
**Eukarya**	Ectosomal	Microvesicles	100–1000	n.d. *	Integrins, selectins, CD40	Intercellular communication, Immunity	[7]
		Apoptotic bodies	1000–5000	1.16–1.28	Annexin V, phosphatidylserine	phagocytosis stimulation	
	Endosomal	Exosomes	30–100	1.13–1.19	Alix, Tsg101, tetraspanins (CD81, CD63, CD9), flotillin	Intercellular communication, Immunity	
**Virus-associated**		Virocell vesicles	n.a. **	n.a. **	virus-specific	Transmission	[26,27]
		Viral Vesicles	n.a. **	n.a. **	virus-specific	Infection support	
		Virion Packaging vesicles	n.a. **	n.a. **	virus-specific	Infection support	
		Virus-Like Particles	n.a. **	n.a. **	virus-specific	Infection support	
		Infectious viral particles	virus-dependent	1.1–1.2 for mammalian virus	virus-specific	Virus propagation, Cellular reprogramming	

n.a. ** depends on carrier vesicle type; n.d. * no accounts found in literature; CD cluster of differentiation; ESCRT endosomal sorting complex required for transport; Tsg101 tumor susceptibility gene 101; virocell vesicles (EVs produced by infected cells, with no viral content present, may however contain elements modified by virus activity); viral vesicles (EVs containing viral nucleic acids); virion-packaging vesicles (EVs containing virions i.e., in Hepatitis A and E); virus-like-particles (replication-incompetent virions). Infection support refers to facilitating of viral transmission by non-infectious virus-associated vesicles; modified from Metzner and Zaruba [24].

The advantages and disadvantages of the different viral vector system are well known, and selection based on application type (e.g., ex vivo vs. in vivo, transient vs. stable) allows one to choose the right viral vector for a wide range of pathologies. The advent of practical, less cumbersome gene correction or editing technologies has even enhanced the impact of VV technology further [35]. Viral vectors are definitely past their coming of age and have reached market authorization status and the clinics with treatments such as Kymriah (Lentiviral), Strimvelis, Zalmoxis (both Retroviral), Luxturna and Zolgensma (all AAV-based) [36]. However, issues regarding the safety and efficiency of viral vectors are still pending. EV-based vectors may prove to be a viable alternative [21].

## 2. Virus-Vesicle-Interplay: Anti- and Proviral Modalities

VIs and EVs share physical, biochemical and functional properties: size, marker distribution (see Table 1) and the capability to migrate between cell types and to re-program the recipient cells as a consequence. EVs are common contaminants in viral preparation (especially of enveloped viruses) and vice versa. EV marker proteins are found on viral particles, as well as viral factors in EVs. Any discriminatory line between EVs and VP may be blurred, and intermediates are described: from virocell vesicles (EVs produced by infected cells, with no viral content present, may however contain elements modified by virus activity), viral vesicles (EVs containing viral nucleic acids), virion-packaging vesicles (EVs containing virions, i.e., in Hepatitis A and E or Herpes Simplex Virus-1) and virus-like-particles (replication-incompetent virions) to fully-infectious virions [26,27,37]. In this review, we will focus on the three main viral vector types: Adeno-, adeno-associated (both non-enveloped) and retro/lentiviral vectors. We will take a closer look at the relationship of the wildtype virus with EVs and how this might impact viral fitness and biomedical use. Both anti- and proviral activities of EVs have been described. While antiviral effects mostly seem to be mediated by EVs from specialized cell types (semen, trophoblast, leukocytes) [1], the general cell populations seem to rather facilitate viral infections. Antiviral responses mostly lead to induction of antiviral states in cellular targets of EVs, while proviral effects mainly fall into two categories: immune evasion and transmission expansion [24].

### 2.1. Non-Enveloped Virus and Virus-Derived Vectors: Adenovirus and Adeno-Associated Virus

Both AV and AAV do not exhibit envelope structures, but are well-studied regarding biochemical, morphological and genetic properties. They have been discussed as vectors for gene delivery for more than four decades and may benefit the most from a little help from EVs by gaining a platform to hide from immunity and acquiring a novel, extended infectivity range. Indeed, while knowledge on the influence of EVs on adenoviral wild type infections is slim, first strategies to exploit EVs for enhancing vector efficacy are under investigation [32,33,34,38]. Further research to elucidate the wt interactions between EVs and AV seems advisable, to better understand (and exploit) physiological and pathological responses as well as avoid unwanted effects from the interactions. Similar to the situation for AV, knowledge on the influence of EVs on wt AAV infections is scarce. Indeed, information on wt parvovirus relationships to EVs is mostly found in the context of research on vector systems [39,40]. For other groups of non-enveloped viruses, such as members of the families *Picornaviridae*, *Reoviridae* and *Papillomaviridae*, interactions with EV are well documented. For these virus families, exocytosis and lysis of infected cells are the preferred options for cell egress. Acquiring a non-classical envelope by hijacking EVs can circumvent this situation. For immune evasion strategies, a different pathway is being followed: instead of using EVs as a decoy (as may be feasible for enveloped virus), particles will be cloaked in cellular membranes, thereby reducing access to the viral antigens. Since the capsid is presented to the exterior and mediates cell attachment, relevant antigens are mostly derived from capsid elements, i.e., those giving rise to neutralizing antibodies. When the capsid is cloaked with the EV membrane, viral antigen display is limited, thus reducing immune responses. *Picornaviridae* (including hepatitis A, poliomyelitis-, rhino-, coxsackie- and foot-and-mouth-disease virus) contain a single ssRNA genome of plus-strand polarity. The viral genome may directly act as mRNA, leading to the translation of all viral proteins. Therefore, viral infection may be established in the presence of a viral genome only, without any additional viral factors being present [24]. For rhino- and poliomyelitis virus, transmission in packages of virion particles (up to thousands) within an EV membrane were documented [41,42]. Such clustering of viruses in vesicles may counteract the fact that infection “fitness” of single virus particles varies considerably, depending on mutations established during replication. Having more particles to infect can reduce the risk of infection with a defective virus. *Reoviridae* (Rotavirus) are also found in lipid-membrane enclosed clusters of particles in stool (similar to *Picornaviridae*). These vesicle-coated clusters survive transmission and, therefore, lead to a high multiplicity of infection (MOI). The data suggested that the lipid-coated clusters are more virulent than free virus [41,43]. Coxsackie, Poliomyelitis and rhinoviral particles use autophagy-related vesiculation for exocytosis [44]. Hepatitis A virus (HAV) in stool, the major route of transmission, is found in the forms of non-enveloped particles. In contrast, host membranes enveloped forms are blood borne and contribute to viremia [45].

### 2.2. Retroviridae-Derived Vectors: Lenti- and Retroviral Vectors (LV/RV)

A common fixture in biomedicine by now, and probably the most successful and promising example of VVs, the members of the family *Retroviridae* seem to have a special relationship with EVs [24,26,27,46] and may even be considered as kind of chimera or hybrid [46]. A fluid spectrum for virus-associated vesicles has been proposed [27]. Interactions are numerous and both pro- and antiviral activities are observed [24,26] (see also Figure 2). In *Retroviridae*, the most profound body of evidence on interactions with EVs has been amassed. However, already carrying an envelope, the help EVs may provide for LV/RV vectors is limited. Mostly, help may be somewhat coincidental: overlaps in biology may lead to an enhancement or optimization of vector production or function [47]. Retroviruses have a very long-lived and intense relationship with their host due to the integration and concomitant latency, which may at least in part explain the close interactions with EVs. While the overlaps between retrovirus and EV biology are astonishing, EV biogenesis seems not to be involved in HIV assembly, since HIV particles bud from the cell membrane (by ectocytosis, rather than exocytosis) [24,48]. However, factors usually found in exosome generation and release are also relevant for retroviral budding [48]. EV proviral effects on the lentiviral model virus HIV include the transfer of accessory proteins such as Vif, which interferes with human restriction factors of the APOBEC family [49] or Nef, which interferes with presentation of many surface receptors, TCR signaling and finally modulates T-cell activation HIV [26,46,50,51]. HIV infection changes the set of miRNAs in cells [52], which appears to be represented in EVs and might have both pro- and antiviral effects. Antiviral effects have been reported for HIV, for EVs derived from human semen and trophoblasts [48]. In addition, prokaryotic EVs derived from symbiotic vaginal lactobacilli inhibit HIV-1 infection [53]. Finally, retroviral sequences present in large quantities in the human genome, termed human endogenous retroviruses (HERVs) [54], are also suggestive, and EVs might play a role in activation or mobility of HERVs [24].

## 3. Discussion—Impact of EV-VI Interplay on Viral Vector Technology

From the relationship of wt virus with extracellular vesicles or, more specifically, exosomes, we have seen that both pro and antiviral responses are possible and, as a consequence, both a beneficial and a detrimental outcome seems possible in relation to VVs. Effects will depend on the source of EVs (patient, cell culture, media, serum or other body fluids) and on therapy modalities (administration route, necessary titers, etc.). Immunity evasion and transmission range/efficacy are the most promising strategies to apply in improving VV applications (see Table 2 for examples).

### 3.1. Technical Issues

Although it may seem trivial, issues regarding the preparation or analysis of VV preparations probably constitute the strongest influence of EVs on VV technology. EVs play a role as contaminants in viral preparation, and especially for enveloped viruses it can be difficult to properly separate the two. However, potentially loosely considered as non-functional VV particles, they would contribute to an unfavourable total-to-infectious particles ratio, a quality parameter for VV preparations [56]. Thus, efficacy of therapies or interventions may be reduced and greater efforts in purification strategies seem advisable. Downstream, analytical procedures may also be hindered by the presence of EVs in VV preps. Marker distribution is overlapping, and viral protein markers may be found on EVs, potentially leading to a misinterpretation of results and—for quantitative results—an overestimation of vector titers. To overcome such issues, single particle analysis of viral particles will help [56]. When gaining information on single particles, ideally in a multi-parametric fashion, distinguishing, quantifying and sorting is possible. Indeed, a process very similar to flow cytometry, which has been termed flow virometry or nanovariant flow cytometry [57], is gaining more and more momentum. Furthermore, a combination of biological and physical parameters may help to unequivocally identify EVs and VIs. Nanoparticle tracking analysis (NTA) and tunable resistive pulse sensing (TRPS) provide information on titer, diameter and zeta potential (a correlate of surface charge) [56,58]. Both techniques have been used for quality control of VVs and vaccine preparations.

### 3.2. EVs Inhibiting VVs

For wildtype virus, inhibitory functions mostly come from the EV signal transduction capabilities (i.e., by inducing an antiviral state in EV recipient cells), an issue that seems less severe since, in most cases, replication incompetent viral vectors are used, allowing only a single round of infection. Therefore, the induction of an antiviral state in potential recipient cells may hardly influence efficacy of therapy. However, the same as for wild type virus would apply in the case of replication-competent viral vectors, which may be used e.g., in oncolytic virus strategies [59]. For non-enveloped viruses, effects considered to be positive for wt virus may actually be detrimental in some cases: cloaking by EVs will hide surface characteristics and potentially change (expand) the tropism range, thus increasing off-target effects. Finding countermeasures for such processes will be difficult. General or localized suppression of vesiculation may be feasible; however, a better understanding of mechanisms is necessary to estimate unwanted effects. Overexpression of targeting factors directed to EV biogenesis may help to alleviate the issue and actually prove to be an asset for EV delivery systems [15,16].

### 3.3. EVs Facilitating VVs

Proviral effects for wt virus mediated by EVs are mostly based on immune evasion or transmission expansion capabilities. VV systems can benefit from such strategies. As suggested by the interactions of wt viruses, such strategies may have evolved most likely naturally as a consequence of virus–host co-evolution.

Immune evasion or manipulation is mostly interesting for in vivo delivery strategies and may indeed be helpful, especially for non-enveloped VVs (AV, AAV). Dominant epitopes can be hidden underneath exosomal cloaks, which will display host, rather than viral, proteins. Examples for this strategy have been tried both in AV and AAV vectors: AAV vectors were found to be more resistant to neutralizing anti-AAV antibodies when associated to EVs [40], and AV-based oncolytic virus was not changed in immunomodulatory capacity when EV associated [32].

Transmission expansion may be of interest for both ex and in vivo approaches. Providing additional anchor points for attachment to the desired target cells may help to increase specificity and efficacy of VV based therapies, especially when using recombinant EVs overexpressing targeting molecules. Indeed, this approach may be extended to immunity modifying strategies by displaying stimulators or inhibitors of specific immune functions. An interesting alternative to genetic manipulation of EV-producing cells is the use of post-exit methods such as Molecular Painting [60,61,62,63,64,65,66] or function-spacer-lipid constructs [67]. In an AV vector system, association of VV to vesicles improved the efficacy of oncolytic virus therapy on tumors with low coxsackie-and-adenovirus (CAR) [34]. In AAV exosome, associated AAV8 vectors were demonstrated to efficiently transduce lymphocytes after systemic delivery [39].

In the case of enveloped viral vectors, help is somewhat limited. Acquiring an envelope structure enabled functions, that may be conferred to non-enveloped viruse by EVs: nonfunctional viral particles may work as an immune decoy (e.g., for Hepatitis B). The envelope may also be used as the primary platform for modification of VVs. However, production capacities may be increased or redirected. Tetraspanins are markers of EVs and have been shown to increase exosome production when over expressed [47]. Interestingly, in an LV vector system, overexpression of the tetraspanin (and exosome marker) CD9 did not increase viral titers but lead to a faster kinetic and an increased efficacy of the vectors [47]. For enveloped viruses, a fine-tuning of production capabilities and functional optimization is feasible by exploiting EV biogenesis, and such strategies may also be adopted for non-enveloped viruses [55].

### 3.4. Perspectives and Outlook

Clinical trials have likely been impacted by EVs. Outcomes may be a tally of pro- and anti-vector effects, probably leading to an overall sub-optimal performance. Measures for a more regulated handling of EVs have been discussed [68,69]. Thus, addressing the balance of EV and VI in VV-based therapies may be a way to optimize the efficacy and safety of VV-mediated biomedical strategies [70]. Due to persisting issues regarding, e.g., immunogenicity of viral vectors, exosomal vectors may prove to be a more secure alternative to classic VVs, depending on a better understanding of EV biogenesis, sub-classes and functionalities. For technical applications, the controlled generation of hybrids or chimeras may prove beneficial by keeping the viral genetics in place [38] but cloak to help with targeting, immune modulation and qualitative and quantitative production parameters. Such chimeras may be implemented by artificial cloaking, i.e., using liposomes [71] or by different methods of coercing vesicles from cells, such as mechanical slicing of cells to generate cellular vesicles (i.e., by forcing through polymer membranes) [34]. Extension of molecular re-design of EV and VI-based biomedical applications by using metabolic engineering and/or synthetic biology may provide a path to develop cellular factories for optimized EV-based delivery applications. On the other hand, viral proteins may be used as “accessory” proteins in EV preparations to manipulate specific target cell functions. Bi-phasic functionalities, i.e., in a “cloak and dagger” fashion, seem promising, and a specifically designed EV shell may protect the vector not only from external damaging sources, but also from premature activation. Expanding on such strategies will, for example, allow combination therapies for cancer treatments by a single delivery event. Of course, potential improvements will depend on the intended use of the vectors. Vaccines based on viral vector platforms may benefit from the use of prokaryotic vesicles as a means to provide adjuvant effects.

We are only at the start and patterns are only beginning to emerge. However, the issues discussed here definitely warrant further research, as a better understanding of the connex between viral replication and extracellular vesicle biology will drastically improve the biomedical capacities of both strategies.

## Figures and Tables

**Figure 1 viruses-13-01238-f001:**
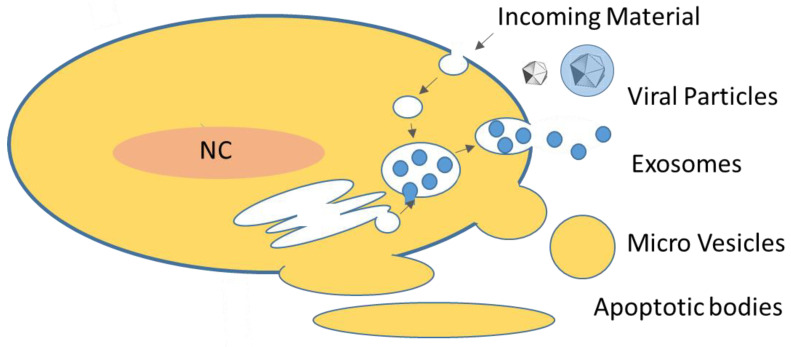
Overview of extracellular vesicles in eukarya: EVs are either produced by membrane shedding (apoptotic vesicles, micro vesicles) or via the endosomal system (a crude schematic of the endosomal system is indicated). Eukaryotic cells produce mainly exosomes, micro vesicles and, under apoptotic conditions, apoptotic vesicles. Exosomes are generated as intra-luminal vesicles (ILVs) in multi-vesicluar bodies (MVB). Incoming material refers to material taken up by cells, e.g., by endocytosis. Also, EVs are entering cells by these mechanisms. For more details see text and Table 1; modified from [24].

**Figure 2 viruses-13-01238-f002:**
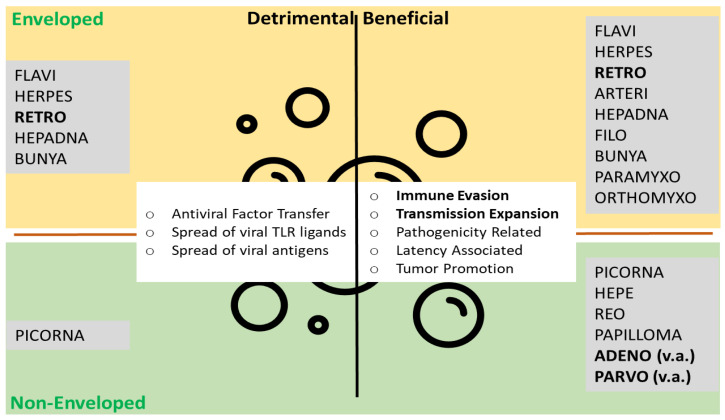
Extracellular vesicles and viral particles. The top lists enveloped viruses associated with EVs, the bottom non-enveloped viruses. On the left, mechanisms detrimental for viruses (antiviral) are listed, on the right, beneficial mechanisms (proviral). Viral families giving rise to the viral vector systems discussed here in bold. Functions most useful for viral vector systems in bold. TLR toll-like receptor, v.a. vector associated. Modified from Nolte-′t Hoen et al. [26].

**Table 2 viruses-13-01238-t002:** Examples for VV-EV interactions.

Figure	Vector	Mechanism	Comment	Reference
Adenoviridae	Ad5D24	CO	PTX oncolytic virus combination	[31]
	Ad5D24	IM, TG		[32]
	Ad5D24	CO	in vivo (murine)	[33]
	n.a.	SF, IM	capsid-free	[38]
	Ad5-P	TM	forced cell vesiculation	[34]
Parvoviridae	AAV2	IM	“vexosomes”	[40]
	AAV1	PR	CD9-overexpression	[55]
	AAV8	TM	lymphocyte transduction	[39]
Retroviridae	LV	PR	CD9-overexpression	[47]

Ad5D24, Ad5-P oncolytic AV strains; AAV1, 2, 8 strains on which AAV vectors are based; CO co-delivery; IM immune modulation; TG targeting; SF safety; TM transmision; PR production.

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
