# Peer review of "On the Relationship of Viral Particles and Extracellular Vesicles: Implications for Viral Vector Technology"

_viruses, 2021, doi:10.3390/v13071238_

Round 1

Reviewer 1 Report

The authors provide a glimpse on the current status of the very difficult and complex field of EVs, VLPs, exosomes,... and their link with viruses. The topic is very interesting and timely to discuss. Still, the current work only briefly touches upon a wide variety of topics: in my opinion this review does not provide the necessary info for people that are new to the field. The scope is maybe too broad to allow a detailed and in focus reviewing on the role of EVs.

Reading through the abstract the authors promise to cover 'Gene therapy vectors', 'extracellular vesicles' and than jump to the fact that 'Viral infections and EV biology overlap in many ways, both with pro- and antiviral outcomes'. In my opinion discussing this will require multiple reviews.   

Major comments

Introduction: EVs are introduced at line22. For non-expert, and since a lot of different definitions are circulating in literature, I reckon it would be best to provide a description of what is considered an EV in this review (size, constituents,…).

Line49: the authors use here the word ‘infect’ to express what happens to EVs when encountering neurons. I do not agree with the wording here. Again, it would be good to provide a better wording for the ways incoming material enters a cell: transduction, transfection, infection, … should be defined in the context of this paper (and preferably in the broader frame of the EV field).

Table1: table should be self-explanatory in my opinion. For the virus-associated extracellular vesicles, please provide more detail to distinguish Virocell vesicles, Viral Vesicles, Virion Packaging vesicles and Virus-Like Particles.

I think it is important to also provide insight that also viral particles are containing EV markers. In addition, it is not clear what ‘infection support’ alludes to.

Line70: VVs do not infect cells, since they are single round. Use of correct wording (here ‘transduction’) is pivotal in this review.

Line112, paragraph 2.1: the title does not cover the content. 2/3 discusses on other non-enveloped viruses

Line162: the authors indicate the close relationship between host and pathogen may in part explain the close interactions with EVs. As an example HIV is put forward, which is in my opinion a rather recent human virus. Is there more info on the interplay in non-human primates (or their cells)?
Endogenous retroviruses on the other hand, are indeed much longer present in our genomes, but these are only very briefly mentioned and not discussed in detail.

Line242: EVs overexpressing targeting molecules: rephrase, EV do not express molecules or proteins, they carry them, no?

Perspectives: “To probably start with a question that is rather moot: In retrospect, have clinical trials been impacted by EVs? While the most likely answer is yes”: the authors claim should be supported by refs and evidence.

In all, the viral vector field is aware of the fact that viral vector preps are not clean. Still, productions used for gene therapeutic applications have nevertheless shown clinical benefit, despite the complexity of the viral vector preps. Studies should indeed be performed and may show that more pure preps are performing better. Still, this is not sure, since several reports also indicated that for example the empty rAAV particles that contaminate the viral vector AAV preps contribute to a more efficient transduction of some tissues.

Minor comments

Line47: ’EVs loaded with siRNA and RVG peptides attached the surface via were 47 used to deliver siRNA to mouse brains intraveneously’: sentence needs rewriting.

Line47: intraveneously should be intravenously.

Line60: why are enveloped viral particles abbreviated as eVI? Since these particles do not replicate, I guess actually “viral vector”, and not “viral” is meant

Line63: the authors refer to ‘evolution and selection of the best adapted means of replication’ I guess, not “optimization”.

Line73: ‘preferential’, ‘conditional’ would probably a better word?

Line80 and 214: virus vector should be viral vector?’

Line97: ‘both lacking an envelope structure’: I would use here non-enveloped viruses (they do not lack it, as if a membrane envelope would be an option)

Line100-102: sentence is difficult to understand

Line110: v.a. vector asociated should be associated

Line155: ‘an especially relationship’ should be ‘a special…’

Line160: ‘coincidentially’ should be ‘coindicental’

Author Response

Comments to Reviewer Comments - Reviewer 1

-The authors would like to thank the reviewer for the detailed assessment and the helpful comments. On hindsight I would agree, that the topic may not be ideal for the form and can provide an entry point to the issue(s), however no in-depth commentary.

The authors provide a glimpse on the current status of the very difficult and complex field of EVs, VLPs, exosomes,... and their link with viruses. The topic is very interesting and timely to discuss. Still, the current work only briefly touches upon a wide variety of topics: in my opinion this review does not provide the necessary info for people that are new to the field. The scope is maybe too broad to allow a detailed and in focus reviewing on the role of EVs.

Reading through the abstract the authors promise to cover 'Gene therapy vectors', 'extracellular vesicles' and than jump to the fact that 'Viral infections and EV biology overlap in many ways, both with pro- and antiviral outcomes'. In my opinion discussing this will require multiple reviews.   

Major comments

Introduction: EVs are introduced at line22. For non-expert, and since a lot of different definitions are circulating in literature, I reckon it would be best to provide a description of what is considered an EV in this review (size, constituents,…).

-We have introduced an earllier definition of EVs and made further reference to table 1for characterization.

Line49: the authors use here the word ‘infect’ to express what happens to EVs when encountering neurons. I do not agree with the wording here. Again, it would be good to provide a better wording for the ways incoming material enters a cell: transduction, transfection, infection, … should be defined in the context of this paper (and preferably in the broader frame of the EV field).

We have replaced the term „infect“ with the more neutral „enter“

Table1: table should be self-explanatory in my opinion. For the virus-associated extracellular vesicles, please provide more detail to distinguish Virocell vesicles, Viral Vesicles, Virion Packaging vesicles and Virus-Like Particles.

-We have added additional information for disambiguation to the table legend.

I think it is important to also provide insight that also viral particles are containing EV markers. In addition, it is not clear what ‘infection support’ alludes to.

-We have clarified the fact, and added an definition of „infection support“.

Line70: VVs do not infect cells, since they are single round. Use of correct wording (here ‘transduction’) is pivotal in this review.

-We have changed the sentence to read „transduction“.

Line112, paragraph 2.1: the title does not cover the content. 2/3 discusses on other non-enveloped viruses.

-We agree on the comment. We have changed the title to „Non-enveloped virus and virus-derived vectors“ to better reflect the content. The text contains a justification of the fact.

Line162: the authors indicate the close relationship between host and pathogen may in part explain the close interactions with EVs. As an example HIV is put forward, which is in my opinion a rather recent human virus. Is there more info on the interplay in non-human primates (or their cells)?
Endogenous retroviruses on the other hand, are indeed much longer present in our genomes, but these are only very briefly mentioned and not discussed in detail.

- We find the evolutionary aspect indicated by the reviewer‘s comment very interesting. Comparing events between long-term (non-human virus, HERVs) and short-term host (HIV) appears to be a good route towards gaining knowledge on the issue. However, we did not want to stray any further.

Line242: EVs overexpressing targeting molecules: rephrase, EV do not express molecules or proteins, they carry them, no?

-Where indicated, we have changed the wording to the more fitting „display“ or „present“

Perspectives: “To probably start with a question that is rather moot: In retrospect, have clinical trials been impacted by EVs? While the most likely answer is yes”: the authors claim should be supported by refs and evidence.

In all, the viral vector field is aware of the fact that viral vector preps are not clean. Still, productions used for gene therapeutic applications have nevertheless shown clinical benefit, despite the complexity of the viral vector preps. Studies should indeed be performed and may show that more pure preps are performing better. Still, this is not sure, since several reports also indicated that for example the empty rAAV particles that contaminate the viral vector AAV preps contribute to a more efficient transduction of some tissues.

-We agree on the issue, introduced extra references and tried to make the point more clear in the Perspectives section.

Minor comments

Line47: ’EVs loaded with siRNA and RVG peptides attached the surface via were 47 used to deliver siRNA to mouse brains intraveneously’: sentence needs rewriting.

-We have changed the sentence.

Line47: intraveneously should be intravenously.

-We have made the correction.

Line60: why are enveloped viral particles abbreviated as eVI? Since these particles do not replicate, I guess actually “viral vector”, and not “viral” is meant

-I assume there is a misunderstanding here. Enveloped virus particles means enveloped virion, rather than enveloped virus-like particle. We have changed enveloped virus particle to enveloped virus to make the sentence more clear.

Line63: the authors refer to ‘evolution and selection of the best adapted means of replication’ I guess, not “optimization”.

-While from a technical viewpoint it constiutes an optimization, this was of course not the intent and not the mechanisms at play. We have tried to make this more clear.

Line73: ‘preferential’, ‘conditional’ would probably a better word?

-While the cause will be the conditions (i.e,. the dysregulation in tumor cells), the outcome is a preference for replication in tumor cells. We have changed this to make it more clear.

Line80 and 214: virus vector should be viral vector?’

-We have changed „virus vector“ to „viral vector“

Line97: ‘both lacking an envelope structure’: I would use here non-enveloped viruses (they do not lack it, as if a membrane envelope would be an option)

-We have changed this.

Line100-102: sentence is difficult to understand

-We have tried to clarify the passage.

Line110: v.a. vector asociated should be associated

-We have changed this.

Line155: ‘an especially relationship’ should be ‘a special…’

-We have corrected this.

Line160: ‘coincidentially’ should be ‘coindicental’

-We have corrected this.

Reviewer 2 Report

Minor Comment

  1. Line 20. ,,Cell host”
  2. Figure 1. The authors have not discussed or mention incoming material or viral particles in the text or figure legend.
  3. Figure 1. is a modified version of a figure from Metzner and Zaruba, 2020 (ref 23). Should mention in the figure legend.
  4. Line 60. Does authors trying to introduce a new term/abbreviation for enveloped viral particles (eVIs) or otherwise need a reference for this abbreviation. None of the references 23-27 use this abbreviation.
  5. Line 93. delete content
  6. Line 97. delete first…… the second view wasn’t mentioned later.
  7. Line 117. wt expand to wild type.
  8. Line 152. RV/LV should be LV/RV
  9. Line 206, 266, and 275. see my comment #4
  10. Line 278. ,,accessory”
  11. Line 279. ,,cloak and dagger”

Author Response

Comments to Reviewer Comments - Reviewer 2

-The authors would like to thank the reviewer for the detailed assessment and the helpful comments. -Replies in italics

    Line 20. ,,Cell host”

-The term „cell dust“ refers to early light-microscop observations of EVs and we have changed the formatting.

    Figure 1. The authors have not discussed or mention incoming material or viral particles in the text or figure legend.

-We have included a reference to the incoming material in the figure legend.

    Figure 1. is a modified version of a figure from Metzner and Zaruba, 2020 (ref 23). Should mention in the figure legend.

-We have included the reference in the figure legend.

    Line 60. Does authors trying to introduce a new term/abbreviation for enveloped viral particles (eVIs) or otherwise need a reference for this abbreviation. None of the references 23-27 use this abbreviation.

- We have tried to make the introduction of the new abbreviation more clear.

    Line 93. delete content

-We have exchanged content with elements

    Line 97. delete first…… the second view wasn’t mentioned later.      

- We have changed the section.

    Line 117. wt expand to wild type.

- We have used the long version.

    Line 152. RV/LV should be LV/RV

-We have corrected this

    Line 206, 266, and 275. see my comment #4

-We have tried to clarify this.

    Line 278. ,,accessory”

-We have corrected this.

    Line 279. ,,cloak and dagger”

-We have corrected this.

Reviewer 3 Report

In this paper, Metzner and Zaruba review interactions among EV biology and adenoviral, adeno-associated and retro/lentiviral vectors, and the implications for viral vector technology and their biomedical applications. This paper is well written and organized and the topic is relevant in the field. However, several minor changes might improve it. Below, I make some considerations (authors’ statements are in bold italics).

General considerations. Viral vectors are being widely used in gene therapy to carry genetic material. Adenovirus (AV) and adeno-associated virus (AAV) vectors are currently among the most used viral vectors in gene therapy. However, several problems with these viral tools have been aroused, mainly the high immune and autoimmune response. Other limitations with viral vectors are that the required tropism is not always easy to find, and their small DNA packaging size may be also a problem. Therefore, alternative systems to deliver exogenous nucleic acids would be desirable. In this context, extracellular vesicles might be new and interesting models to deliver foreign genetic material to cells. Orefice (2020), among others, has reviewed this topic recently. In my opinion, the manuscript should state more clearly this background in the introduction or in the abstract, in addition to the relevant relationship between EVs and VV that have been described in the text, (such as, for instance, the inhibitory or proviral functions of EVs).

LINE 22. “Reviews on the topic are numerous [1-9].”

I consider that a relevant reference should be included here:

J Extracell Vesicles. 2015 May 14;4:27066. doi: 10.3402/jev.v4.27066. Biological properties of extracellular vesicles and their physiological functions. María Yáñez-Mó et al 2015.”

LINE 22. “Extracellular vesicles (EVs) are produced from eukaryotic as well as prokaryotic and archaeal sources”.

The sentence “prokaryotic and archaeal” is redundant, since archaea are prokariots, I consider that “bacterial and archaeal” would be more accurate

LINE 72. “such as chimeric antigen receptor (CAR) based immune-therapies [28].”

More references would be advisable, such as:

June, C. H., & Sadelain, M. (2018). Chimeric Antigen Receptor Therapy. New England Journal of Medicine, 379(1), 64–73. doi:10.1056/nejmra1706169

LINE 94. “virion-packaging vesicles (EVs containing virions i.e. in Hepatitis A and E)

HSV-1 is another virus that has been observed inside EVs:

J Virol. 2018 Apr 27;92(10):e00088-18. doi: 10.1128/JVI.00088-18. Role of Microvesicles in the Spread of Herpes Simplex Virus 1 in Oligodendrocytic Cells. Bello-Morales et al.

Author Response

Comments to Reviewer Comments - Reviewer 3

-The authors would like to thank the reviewer for the detailed assessment and the helpful comments.

In this paper, Metzner and Zaruba review interactions among EV biology and adenoviral, adeno-associated and retro/lentiviral vectors, and the implications for viral vector technology and their biomedical applications. This paper is well written and organized and the topic is relevant in the field. However, several minor changes might improve it. Below, I make some considerations (authors’ statements are in bold italics).

General considerations. Viral vectors are being widely used in gene therapy to carry genetic material. Adenovirus (AV) and adeno-associated virus (AAV) vectors are currently among the most used viral vectors in gene therapy. However, several problems with these viral tools have been aroused, mainly the high immune and autoimmune response. Other limitations with viral vectors are that the required tropism is not always easy to find, and their small DNA packaging size may be also a problem. Therefore, alternative systems to deliver exogenous nucleic acids would be desirable. In this context, extracellular vesicles might be new and interesting models to deliver foreign genetic material to cells. Orefice (2020), among others, has reviewed this topic recently. In my opinion, the manuscript should state more clearly this background in the introduction or in the abstract, in addition to the relevant relationship between EVs and VV that have been described in the text, (such as, for instance, the inhibitory or proviral functions of EVs).

We have included a statement in the introduction/discussion section.

LINE 22. “Reviews on the topic are numerous [1-9].”

I consider that a relevant reference should be included here:

J Extracell Vesicles. 2015 May 14;4:27066. doi: 10.3402/jev.v4.27066. Biological properties of extracellular vesicles and their physiological functions. María Yáñez-Mó et al 2015.”

-We will include the reference.

LINE 22. “Extracellular vesicles (EVs) are produced from eukaryotic as well as prokaryotic and archaeal sources”.

The sentence “prokaryotic and archaeal” is redundant, since archaea are prokariots, I consider that “bacterial and archaeal” would be more accurate

-We have used the more precise wording now.

LINE 72. “such as chimeric antigen receptor (CAR) based immune-therapies [28].”

More references would be advisable, such as:

June, C. H., & Sadelain, M. (2018). Chimeric Antigen Receptor Therapy. New England Journal of Medicine, 379(1), 64–73. doi:10.1056/nejmra1706169

-We have included the extra reference.

LINE 94. “virion-packaging vesicles (EVs containing virions i.e. in Hepatitis A and E)

HSV-1 is another virus that has been observed inside EVs:

J Virol. 2018 Apr 27;92(10):e00088-18. doi: 10.1128/JVI.00088-18. Role of Microvesicles in the Spread of Herpes Simplex Virus 1 in Oligodendrocytic Cells. Bello-Morales et al.

-We have included the reference and made a notion of the fact in the text.